# Changes in Mental Health, Emotional Distress, and Substance Use Affecting Women Experiencing Violence and Their Service Providers during COVID-19 in a U.S. Southern State

**DOI:** 10.3390/ijerph20042896

**Published:** 2023-02-07

**Authors:** April Schweinhart, Camila Aramburú, Rachel Bauer, Ashley Simons-Rudolph, Katharine Atwood, Winnie Kavulani Luseno

**Affiliations:** Pacific Institute for Research and Evaluation, 4061 Powder Mill Road, Suite 350, Beltsville, MD 20705, USA

**Keywords:** intimate partner violence, mental health, substance use, COVID-19, power and control

## Abstract

Research conducted during the COVID-19 pandemic has revealed many unintended consequences of mandated safety precautions, including increased perpetration of intimate partner violence (IPV), increases in substance use, and worsening mental health conditions. We conducted a repeated, cross-sectional survey of survivors of IPV, a longitudinal survey of service providers working in an IPV shelter, and interviews with both. We conducted surveys at the beginning of the pandemic and nearly half a year later to assess mental health and, for clients, substance use. Results showed that two small samples of survivors living in the shelter in 2020 and 2021 experienced both mental health decline and increased use of substances. Qualitative data from in-depth interviews suggest that COVID-19-related restrictions mirrored survivors’ experiences of power and control in violent relationships. Further, IPV service providers—essential workers during COVID-19—experienced stress associated with reports of burnout and mental fatigue. This study suggests that community-based organizations can help mitigate the impacts of COVID-19 on survivors of IPV but should avoid adding additional work for staff as service providers experienced mental and emotional stress.

## 1. Introduction

Soon after stay-at-home orders for the COVID-19 pandemic were issued, researchers began studying how these orders, precautions, and the pandemic affected individuals. Many of these studies focused on the effects of the pandemic on mental health and substance use in the general population and found that the unintended consequences of COVID-19-related public health orders include psychological distress, fear of infection, insufficient resources, and income insecurity [1,2,3,4]. For example, adults were three times more likely to meet the criteria for depressive and/or anxiety disorders in April and May 2020 than they were in 2019 [5] and eight times more likely to fit the criteria for serious mental distress in 2020 than they were in 2018 [6]. In addition to the increase in mental health conditions, data collected in 2020 show an increase in substance use. More specifically, 13.3% of adults surveyed in 2020 reported that they had started or increased their substance use to cope with stress or emotions related to the pandemic [7]. Impacts on mental health and coping behaviors due to the pandemic were found across all demographic groups, but most significantly among younger people [5,6,7,8,9], female-identifying individuals [8,9], persons with low financial assets [10], persons with children in the household [5], and those identifying as black or Hispanic [7,9]. Additionally, strong predictors of changes to mental health included: pre-pandemic psychiatric diagnoses [7,8,11,12], past victimization [11], job and/or wage loss or high COVID-19 financial stressors [10,11,12], smoking [12], a higher consumption of alcohol or cannabis [8], and worry about contracting COVID-19 [12]. 

The effects of COVID-19 on the general population may be compounded for those experiencing violence, as research shows that well-documented co-morbidities of violence include substance misuse and mental health concerns. According to one review, over half of women seeking treatment for mental health had experienced IPV [13,14], and IPV has been associated with depression, PTSD, anxiety, dysthymia, and phobias [15,16]. Research shows that having a diagnosis of serious mental illness indicates a greater likelihood of experiencing violence, but victimization also indicates a greater likelihood of having a serious illness diagnosed [13,14,17,18,19]. Further, IPV and substance use have a complicated, bi-directional relationship, as substance use might be a response to trauma and/or increase the likelihood of trauma occurring [20,21,22]. 

Many have already investigated the relationship between COVID-19 and increasing rates of IPV perpetration that is physical, sexual, or consequential of perpetrator substance use [1,23,24,25,26,27,28,29]. Not only have rates of IPV increased, but some researchers have also suggested that violence during COVID-19 may have been more severe, as indicated by the types of injuries sustained [30]. For instance, sexual and physical (as opposed to, for example, emotional) violence increased [29,31]. In a 2021 online survey of 53 individuals experiencing safety concerns related to IPV, more than 40% of the participants reported decreases in safety [32]. Two other studies found that individuals experiencing IPV reported more stress than those in relationships without IPV and that COVID-19 caused loneliness among survivors of IPV [33,34]. A fear of COVID-19 infection was also a reported barrier to seeking help for IPV during the pandemic [35,36]. Authors also cite an underreporting of violence as well as a lack of services available as affecting the impacts of increases in violence perpetration during the pandemic [23,28,37]. These research findings prompted calls to further integrate services across intersecting domains, such as mental healthcare and substance use treatment, to better reach individuals experiencing violence [24,28,31,35,38,39]. 

Public health directives, such as wearing personal protective equipment (PPE), stay-at-home orders, and social distancing, were essential to mitigating the spread of COVID-19, especially in communal living spaces such as IPV shelters [40]. However, less is understood about how the pandemic and related public health orders affected survivors of IPV. As an example, Welfare-Wilson et al. [41] reported on their own psychological distress from wearing masks as survivors of IPV, as it served as a physical reminder of the abuse they had endured. Fishere et al. [42] examined populations in Egypt, Germany, and Italy with a history of trauma and determined that those who had experienced trauma were more at risk for the negative mental health impacts of the pandemic than participants without past exposure to trauma (most instances of IPV would be considered traumatic by those experiencing them). Often, social isolation is a tactic perpetrators of violence use to separate victims from resources, family, and friends and may be re-experienced by survivors when their government directs them to self-isolate [43]. Further, income insecurity due to employment changes (e.g., unemployment or reduced work hours) during the pandemic was also found to be an instigator of stress in women [3].

While the literature on the impacts of the pandemic on IPV survivors is limited, evidence has been published on the COVID-19 impacts of another vulnerable population, survivors of war. Both during war and during the pandemic, communities have witnessed dramatic decreases in social and tangible support systems [44]. Changes in access to social services can exacerbate the problems of already underserved populations [4,45,46]. Further, Stolow et al. [47] touched on the negative consequences of officials using fear-based approaches to gain compliance from their communities. These approaches can amplify an already stress-inducing event, such as the pandemic, in addition to causing unfavorable reactions and driving a wedge between officials and the community they serve. Survivors of the Holocaust may be retraumatized by public health restrictions such as the lockdown, as they are reminiscent of the wartime controlling of movement, resources, and information [44]. Similarly, survivors of the Bosnian and Herzegovinian genocide reported greater PTSD symptoms and generally greater impacts of the pandemic [48]. Power, manipulation, and fear are all tactics used by perpetrators of violence to control their victims, as is described in Duluth’s Model of Power and Control [49], which we have adapted to consider the connection with the impacts of COVID-19 [50]. In sum, these findings seem to suggest public heath orders implemented during the pandemic may retraumatize survivors of violence by triggering previous trauma or replicating experiences and situations similar to violence.

Importantly, the noted decreases in available services for IPV could be related to COVID-19 effects on service providers, including burnout, mental distress, and high turnover rates. Generally, providers labeled as essential workers experienced greater stress, a lack of access to necessary resources (PPE), and greater burnout during COVID-19 than before [40,51,52,53]. Many social service providers indicated they were unprepared for the professional challenges of COVID-19 [53] and were impacted personally by the pandemic’s primary consequences, such as not having enough access to food or healthcare and becoming ill, as well as secondary consequences such as stress [32,53]. Professional support was noted as key for continuing to provide services [51,52,54]. Many providers also experienced increased caseloads, and some researchers found that the additional stress and workloads affected the quality of services provided [55]. The impact of the pandemic on providers of care is important to consider when researching its impact on clients seeking care, because the quality of care may be affected by shared environmental circumstances.

The intersectional relationships between IPV, substance misuse, and mental illness, coupled with an increase in all three during the pandemic, led us to examine changes in substance use and mental health concerns among IPV survivors and service providers. We hypothesized that survivors of violence may be experiencing greater mental health concerns and greater impacts of violence during the pandemic and that service providers working with survivors may be experiencing negative effects of stress similar to those noted among other helping professions. Further, the increased stress in providers may be passed on to survivors if the quality of care provided is affected by the pandemic. The motivation for this study included the need to notify service providers of survivors’ mental health and substance use changes so they could better address these concerns with their clients [23,28,31,32] as well as to notify IPV service-providing organizations of the need to support their staff. As part of a larger study examining the intersection of HIV, substance misuse, and violence, we asked survivors living in a domestic violence shelter in a Southern U.S. State and the IPV service providers working in the same shelter about their COVID-19-related experiences. We also conducted in-depth interviews with the same populations to further explain any changes in self-reported mental, physical, or substance use behaviors.

## 2. Materials and Methods

Our research team used a mixed-methods approach to studying changes in mood and behavior related to the COVID-19 pandemic among survivors residing in an IPV shelter as well as the service-providing staff at the same organization. Quantitative data were collected using a repeated cross-sectional study design for clients and a longitudinal design for service providers. We sampled clients and service providers via convenience sampling and voluntary response sampling. In-depth interviews were conducted to collect qualitative data. All interview and survey questions, summaries of survey data, and interview themes were reviewed by a community advisory board, consisting of seven women with lived violence and/or substance use experience. Consent was obtained for all staff, clients, and community advisory board members to participate in data collection and interpretation efforts.

Original data collection protocols involved in-person, one-on-one interviews and client survey recruitment by the study team. However, the COVID-19 pandemic disrupted in-person data collection efforts. Following state-wide social distancing mandates, the data collection team, in collaboration with IPV shelter staff, developed new protocols for remote data collection methods, specifically the use of online survey platforms (Qualtrics) and web-based conferencing tools (Microsoft Teams) for conducting staff surveys, staff and client in-depth interviews, and community advisory board meetings. All study protocols and instruments were approved by the Pacific Institute for Research and Evaluation’s Institutional Review Board (IRB).

### 2.1. Quantitative Data Collection (Clients)

Individuals living through violence and residing at the IPV shelter (clients) were recruited to participate in two rounds of surveys and one in-depth interview. The client surveys included measures near the beginning of the COVID-19 lock down (summer 2020) and at a second time about six months later (spring 2021). These measures asked participants about the impacts and challenges of the COVID-19 pandemic and about changes in their substance use since the pandemic began.

#### 2.1.1. Instrument Development for Clients

Our COVID-19 instruments were based on measures available in the published literature. Nearly all of these published measures have been validated and/or are accessible via the PhenX Toolkit [56,57,58,59,60,61]. Unique items across instruments were used to identify population-relevant COVID-19 stressors and to measure the impact of COVID-19 on clients’ health, well-being, and behaviors at time 1 and time 2. Participants were first asked to indicate (yes = 1, left blank = 0) if they had experienced any of the following COVID-19 stressors:A COVID-19 diagnosis;Worry/anxiety about being infected;Changes in physical health;Not having enough basic supplies (e.g., food, water, medications, or a place to stay);Children experiencing distress;Interpersonal conflict with family members or loved ones;Feeling more depression;Feeling more loneliness;Feeling less safe;Feeling more abused (physically, emotionally, or psychologically).

Participants were also given the option to indicate, “No changes to my life or behavior.” For each item above, if individuals reported that they had experienced it, they were then asked to indicate on a 5-point scale (0 = not at all, 1 = a little, 2 = somewhat, 3 = more than some, and 4 = greatly/a lot) how much each experience affected them. Participants were also asked if they engaged in any of the following activities, feelings, or behaviors to overcome COVID-19 stressors (yes = 1, left blank = 0; also developed from the review of COVID-19 instruments):Feeling grateful for the break from usual activities;Connecting with others, including talking with people you trust about your concerns and how you are feeling;Needing more/less sleep or other changes to your normal sleep pattern;Taking breaks from watching, reading, or listening to news stories, including social media;Healthy behaviors such as trying to eat healthy, well-balanced meals, exercising regularly, getting plenty of sleep, or avoiding alcohol and drugs;Engaging in more family activities;Talking with a mental health care provider.

Participants were also given the option to indicate, “None/no changes.” If individuals indicated yes to any of the coping mechanisms listed, they were then asked to indicate on a 5-point scale (0 = not at all, 1 = a little, 2 = somewhat, 3 = more than some, and 4 = greatly/a lot) how much each activity, feeling, or behavior helped them cope.

Lastly, clients were asked to report (yes = 1, left blank = 0) if they currently use tobacco (e.g., smoking cigarettes, e-cigs, vapes, or chewing tobacco), alcohol, and/or illicit or illegal substances (e.g., marijuana, prescription drugs not prescribed to you, or meth). If individuals reported any use, they were then asked to indicate how much their use of the specified substance changed since the COVID-19 pandemic using a 5-point scale (1 = went down a lot, 2 = went down slightly, 3 = stayed the same, 4 = went up slightly, and 5 = went up a lot). The following individual characteristics were also collected via the survey: marital status, age, number of children, gender, level of education, health insurance, employment, and race/ethnicity. Other survey measures were related to HIV knowledge, self-efficacy, and behavior, but they are not the subject of this study.

#### 2.1.2. Survey Administration with Clients

Shelter staff invited residents to complete the 30 min, anonymous paper-and-pencil surveys between 15 July and 21 August 2020 and between 1 March and 13 April 2021. Each recruitment period included completely distinct sets of individuals because residential shelter clients changed between 2020 and 2021. Only female clients that were 18 years or older and residing in the domestic violence shelter during the time of recruitment were eligible to participate. The study team developed flyers, advertisements, and scripts that detailed the study purpose and protocols to assist staff in recruiting participants. If a resident indicated an interest in participating, staff distributed a survey packet including study information and consent forms, the survey, a list of resources for IPV, HIV, and substance use services, and an envelope to seal their survey in once completed.

Shelter residents indicated their consent by completing the survey and returning the sealed envelope to a staff member. Once their survey was completed and returned to the designated staff member, the client received a $15 gift card to a local grocery store. Sealed envelopes with completed surveys were picked up at the shelter weekly by a member of the research team. Survey responses were then entered into an online, password-protected survey development software (Qualtrics) by a trained research associate. Responses were exported to an excel document and merged using SPSS Statistics 27 for analysis.

### 2.2. Qualitative Data Collection (Clients)

In-depth interviews with clients were held during the fall of 2020. A member of the data collection team that was trained in human-subjects research as well as IPV counseling conducted the interviews to gain insight into clients’ thoughts on HIV, HIV testing, and behaviors related to COVID-19.

#### 2.2.1. Interview Guide for Clients

To ensure participants’ safety from perpetrators or others who may have been able to overhear interviews, all participants were residing in the shelter at the time of the interview and were escorted to and from a private computer room by shelter staff members. All interviews were conducted using end-to-end encrypted software (Microsoft Teams 1.6) to ensure confidentiality. Our research team developed an interview guide based on the initial responses from quantitative measures indicating that the pandemic was affecting individuals residing in the shelter. The interviewer asked the following questions and prompts:Since COVID-19 is on our minds, I’d like to start by asking about how COVID-19 has or has not impacted the ways that people you know think about their health. How has the virus impacted the way people think about health risks like IPV, HIV, and SUD?How has COVID-19 affected people that you know in their ability to access:
IPV services;Substance use services;Services related to your sexual health (exams, testing, etc.);Other health care services?
Is there anything else you would like to tell me about how COVID-19 has impacted people with interpersonal violence in their lives?

Other questions were asked about HIV and risks of violence but are not reported here.

#### 2.2.2. Interview Implementation with Clients

Individuals recruited from the same IPV shelter were recruited for interviews by shelter staff, as described in the preceding section, between 31 August 2020 and 4 November 2020. To ensure client safety and reduce the chances of retraumatization, only individuals residing in the shelter and demonstrating emotional stability at the time of recruitment (as determined by shelter staff) were invited to participate. Additionally, only individuals that were 18 years or older and identifying as female were eligible to participate. Shelter staff distributed interview packets to interested participants. Interview packets included a copy of the consent form, examples of the interview questions, and a resource list with contact numbers for local medical and violence service providers. Interviewees may or may not have also participated in client surveys. Identifying information was not cross checked.

Prior to beginning each interview, the interviewer asked each participant to confirm that they were in a safe, private location, reviewed the consent form, and obtained verbal consent. The interviewer also asked participants if they felt comfortable continuing the interview and if their privacy level changed throughout the interview. Interviews took between 30 and 60 min to complete and explored the impacts of COVID-19, including personal experiences, changes in access to services, and effects of COVID-19 in the context of IPV, HIV, and substance use. After each interview, the participant received a $30 gift card to a local grocery store via email.

The interviewer was trained to identify, respond to, and assess the client’s distress; determine the next steps (to conclude or continue the interview); and follow up with IPV service-providing staff to report any distressing incidents (none were reported). In-depth interviews were conducted, recorded, and transcribed using Microsoft Teams. The transcribed files were uploaded to Dedoose, a mixed-methods software (version 8.3.43), to be coded and analyzed.

### 2.3. Quantitative Data Collection (Staff)

Staff working in the shelter where clients were residing (staff) were invited to participate in a longitudinal survey (i.e., two rounds of data collection) and one close-out interview. These staff surveys included baseline (spring 2020) and follow-up (winter 2021) measures asking participants about the impacts and challenges of the COVID-19 pandemic.

#### 2.3.1. Instrument Development for Staff

Similar to the instruments used with clients, dichotomous and Likert-scale survey items were created based on a mutually exclusive review of available COVID-19 instruments [56,57,58,59,60,61]. A unique ID was created for each participant to be able to assess their individual changes over time. Participants were first asked to indicate (yes = 1, left blank = 0) if they had experienced any of the following COVID-19 stressors and workplace challenges:Worry/anxiety about being infected;Changes in physical health;Challenges in workplace due to social distancing;Challenges in workplace due to loss of childcare;Challenges in workplace due to technology issues (loss of internet, lack of necessary equipment);Challenges in workplace due to services normally provided impeded by the precautions;Challenges in workplace due to changes to client mental or physical health;Challenges in workplace due to changes to how services are provided to clients.

For each item above, if individuals reported they had experienced it or any workplace challenge, they were then asked to indicate on a 5-point scale (0 = not at all to 4 = greatly/a lot) how much each experience affected them. If participants experienced changes in how they provided services to clients at the IPV shelter, they were also asked to indicate on a 5-point scale (0 = not at all to 4 = greatly/a lot) how confident they felt in administering services in the new way. Additionally, staff were given the option to indicate “yes” or “no” on whether they had experienced other changes to their lives or behaviors due to COVID-19. If a participant indicated “yes,” they were encouraged to then list the additional changes to their life or behavior that they had experienced as a result of COVID-19. Other items in the survey asked about HIV knowledge and having the self-confidence to ask clients about sexual safety planning, but they are not the subject of this study.

#### 2.3.2. Survey Administration to Staff

All shelter staff, including, but not limited to, advocates, therapists, group leaders, supervisors/directors, and service coordinators, were invited by the study team and encouraged by leadership at the shelter to complete the 25 min, confidential, online baseline survey between 1 June and 30 July 2020 and the follow-up survey between 18 January and 9 March 2021. The study team and leadership introduced the purpose of the survey during an all-staff meeting. Afterward, an email was sent to each staff member reintroducing the study and inviting them to participate, followed by personalized links to the survey developed by Qualtrics, and distributed to each of the staff via email. Staff received three email reminders during the survey administration period. To enhance staff participation, the study team followed up with department leads to encourage their team members to complete the survey, and staff who completed the survey were entered into a drawing to win one of four $50 Visa gift cards. The winners were randomly selected using Excel. Survey responses were exported to an excel document and merged using SPSS for analysis.

### 2.4. Qualitative Data Collection (Staff)

In-depth interviews with staff were held as project close-out interviews during the summer of 2021. Three members of our team that were trained in data collection with human subjects conducted the in-depth interviews to gain insight into the staff’s thoughts on implemented program activities regarding HIV, HIV testing, and behaviors related to COVID-19.

#### 2.4.1. Interview Guide for Staff

All interviews were conducted using end-to-end encrypted software (Microsoft Teams 1.6) to ensure confidentiality. Our research team developed an interview guide based on responses from quantitative measures and activities conducted as part of the larger project. The interviewers, all qualitative methods-trained researchers, asked the following questions and prompts:While we are really proud of our work together on this project, one thing we found is that staff report that they still struggle with conversations with clients about substance use, HIV, and IPV. What are your thoughts about this? (Probe: What do you think might be keeping staff (or you) from having conversations about substance use, HIV, IPV?)Some IPV shelter clients have taken advantage of the opportunity to get a free, confidential HIV test in shelter. Others have not. Some of these clients have chosen to go elsewhere to get tested. Why do you think women are not getting tested in shelter?Thinking back, what did you appreciate about this project? (Activities included training on HIV, training on how to talk to clients about HIV and make sexual safety plans, new protocols for HIV testing in shelter and for non-residential clients, and surveys of staff and clients).What would you have done differently?Anything else that we should know as we wrap up the project and write up our lessons learned for other IPV advocate service agencies?

#### 2.4.2. Interview Implementation with Staff

Staff from the same IPV shelter were recruited for interviews via emails sent from project researchers between 1 October and 4 November 2021 (interviews took place from 8 October to 4 November). Prior to beginning each interview, the interviewer reviewed the consent form and obtained verbal consent from the participant. The interviews were recorded and transcribed for analysis. After each interview, the participant was emailed a $10 gift card. Although none of the interview questions for staff asked anything specifically about experiences with COVID-19, some staff responses were relevant as explained in Section 3.

### 2.5. Data Analysis

#### 2.5.1. Survey Data Analysis

A trained statistician cleaned the data and removed any identifying information before conducting descriptive analyses on the survey responses. Many participants seemed confused by the presentation of the questions that asked them to both indicate if an experience had happened and then rate how much it had affected them. To correct for any confusion, frequencies of experiences include any response of “yes,” indicating the experience happened, and any response to the level of affectedness that was greater than zero. That is, if a participant did not respond to the item asking if the event happened, but did rate it as having affected them, that item was counted as a positive response to the experience having happened.

Our research team discussed the descriptive results, including the means, standard deviations, and frequencies of responses, before determining final analysis plans. Further analyses included conducting paired and independent tests for differences in means (Student’s *t*-test), correlations between survey response items (Spearman’s r-test), and non-parametric measures of sample differences (Mann–Whitney U-test).

#### 2.5.2. Interview and Open-Ended Response Data Analysis

The qualitative data from the client interviews were segmented to develop a coding scheme using an iterative and inductive approach. After the initial coding scheme was developed, the data were exported by the software and catalogued in a codebook. Another member of the project team completed a preliminary review of the coding scheme and findings to further organize and hone themes into more descriptive categories as well as to ensure uniformity across all aspects of the analysis. Both primary analysts conducted additional reviews of the thematic analysis to reach a consensus on the main themes and to identify significant findings. After the initial review, the findings were presented to the larger team for discussion and review to finalize themes. Findings were also shared with the project’s community advisory board, who championed the idea and helped to construct a conceptual model that aligns with Duluth’s Model Power and Control Wheel [49] (see Results). These findings were first presented at the 14th Annual InWomen’s Conference [50].

Staff interviews were similarly analyzed using a grounded theory approach [62] by one reviewer that was trained in qualitative methodology, and codes were reviewed by the project director (first author). Like those of client interviews, the results of staff interviews were shared with the community advisory board. Open-ended responses from staff surveys were iteratively reviewed and segmented into themes and thematically coded using a grounded theory approach as well.

## 3. Results

### 3.1. Quantitative Results (Clients)

During the first data collection period, a total of 19 clients completed the surveys. A different set of 15 clients completed the same surveys 5–7 months later, at time 2. The demographics for clients who participated in the surveys (time 1 and time 2) are listed in Table 1. The frequency and percent response rates for each COVID-19 experience, as well as the average response scores for how much each item affected clients, are listed in Table 2. Some of the frequency responses between clients selecting a COVID-19 experience and rating their level of affect may differ due to missing data. No clients at time 1 and only two clients at time 2 reported that they had experienced “no changes to [their] life or behavior.”

All items related to COVID-19 were experienced by a majority of the participants at time 1, including 68% indicating that they had experienced a COVID-19 diagnosis and/or had changes to their physical health (Table 2). By time 2, no clients indicated experiencing COVID-19 diagnoses, and around one-third indicated worry about being infected (33%), changes in physical health (33%), not having enough basic supplies (40%), or their children experiencing distress (33%). However, although the percentage scores decreased for the following items from time 1 to time 2, a majority of clients still indicated experiencing interpersonal conflict (53%), feeling more depression and loneliness (60%), feeling less safe (60%), and feeling more abused (67%). Moreover, average scores for how clients were affected by these last five items increased between clients surveyed at time 1 and those surveyed at time 2. To determine if these increases were significant, we conducted an independent-samples *t*-test on the clients’ reported affectedness for each item, using Bonferroni correction for multiple analyses. As captured in Table 2, none of the changes from time 1 to time 2 were significant, indicating that clients at each time reported the same level of affectedness for these experiences.

Table 2 also depicts results of the coping strategies clients used. At time 1, only two clients (11%) indicated that they had experienced none of these coping strategies, and at time 2, zero clients indicated that they had not used any of these coping strategies. Like with COVID-19 experiences, most clients reported using all the listed coping strategies at time 1, and fewer indicated using these strategies 5–7 months later; exceptions include “feeling grateful for the break from usual activities” and “engaging in more family activities,” both of which increased in prevalence by time 2. We also conducted independent-samples *t*-tests for how much each of the coping strategies affected clients and found no differences between time 1 and time 2 (corrected alpha of 0.006).

Table 3 shows changes in clients’ substance use from time 1 to time 2. Most clients at time 1 indicated that their use of alcohol (90%), illegal substances (58%), and tobacco (68%) had changed. In the interviews 5–7 months later, fewer clients indicated that their use of alcohol (73%), illegal substances (27%), and tobacco (33%) had changed. Earlier in the pandemic (time 1), clients, on average, reported that their substance use for all three categories had increased (scores > 3). By 5–7 months later, clients reported using less alcohol and tobacco (scores < 3), but still more illegal drugs since the start of COVID-19 as well as since time 1, although these reported changes in the amount of use were non-significant (Table 3).

We also conducted Spearman’s rank-order correlations to determine if reports of survivors’ experiences of more loneliness, depression, and abuse were related to reports of greater substance use. Again, we used a Bonferroni correction (alpha rate of 0.006) as a conservative correction to account for running multiple correlations. At time 1, feeling more loneliness was associated with feeling more depression and feeling more abused. At time 2, the relationships changed such that depression was associated with feeling more loneliness, less safe, and more abused. Table 4 shows the complete results of these analyses.

### 3.2. Qualitative Results (Clients)

Interviews were conducted with six clients receiving services in the IPV resident shelter. We conducted a thematic coding of the client interviews, which yielded three primary themes related to COVID-19: (1) IPV survivors experienced impacts of COVID-19 on their mental and physical health, (2) survivors shared the effects of stay-at-home orders on support system/services access and financial security, and (3) COVID-19-related experiences were retraumatizing for clients because they mirrored how abusers use power and control to perpetrate aspects of IPV. These identified themes were mapped onto most of the elements of the Duluth Model Power and Control Wheel of IPV [49] presented in Figure 1 and present a substantial overlap. Figure 2 depicts the relationship between COVID-19 experiences reported by survivors interviewed in this study and the traditional elements of power and control used by violent perpetrators, as well as exemplary quotes from survivors relating to each element.

#### 3.2.1. COVID-19 Impacts on Clients’ Physical and Mental Health

Participants reported that their physical health was at risk during the pandemic given the increased risk of infection of COVID-19 in addition to the postponing of needed healthcare as a result of navigating care in an overburdened healthcare system. Medical professionals’ focus on COVID-19 infections also meant there was less focus on other health needs, such as HIV and IPV care, according to participants. Broadly, clients described increased stress levels relative to changes in income, access to services and support systems, and worry and uncertainty about COVID-19 testing and screening. Further, participants shared the compounding effects of having a history of trauma with the emotional impacts of the pandemic. One participant described this by saying, “People feel like they are not going to make it [emotionally]. There is a heaviness like a burden.” An increased engagement in negative coping mechanisms, such as substance use, and a decrease in resiliency was also identified.

#### 3.2.2. COVID-19 Limited Access to Support Systems/Services and Financial Security

Like a violent partner, COVID-19 restrictions limited access to supportive systems such as family and friends because guidance included avoiding contact with anyone outside of one’s household. While this separation could mitigate the spread of infection, clients indicated that it led to stress, a loss of social connectedness and feelings of support, and other negative mental health issues. Additionally, clients were affected by COVID-19 regulations around health care providers’ offices, including longer wait times, providers not accepting new patients, offices disallowing supportive health advocates, stifling personal protective equipment requirements, the burden of following public health guidelines (e.g., having to shout personal information through a plastic screen), and a fear of COVID-19 infection. Participants also shared challenges related to sexual healthcare, safe-sex supplies, and contraception choices, all of which can also be made difficult to obtain or use appropriately when in a violent relationship. Financial instability was also posed given changes in employment status.

#### 3.2.3. Clients Were Retraumatized by COVID-19 Restrictions

Interviews suggested the pattern of COVID-19-related health directives was similar to how perpetrators of violence use power to control their partners. Our findings suggest that the virus, in addition to government-sanctioned control over bodily autonomy (mask mandates) and restrictions to movement, was retraumatizing for survivors. Further, participants indicated that wearing personal protective equipment made individuals appear menacing. Power differentials (privilege) between clients and their providers were also described as being exacerbated given changes to healthcare access and strict protocols for accessing services. Feelings of lost agency imposed by the shelter’s policies were also described by a participant who was required to have a tuberculosis test before being allowed to reside in the shelter.

### 3.3. Quantitative Results (Staff)

During the four-week baseline data collection period, a total of 59 staff members participated in the survey. In the follow-up survey, 40 staff members participated. Of those 40, 35 also took the baseline survey. The demographics of all staff who participated in the surveys (baseline and follow up) are listed in Table 5.

The frequency and percentage response rates for each COVID-19 experience, as well as the average response scores for how much each item affected the staff, are listed in Table 6. Some of the frequency responses between staff selecting a COVID-19 experience and rating their level of affect may differ due to missing data. Only two staff at baseline and no staff at follow up reported that they had experienced “none of the above” changes to their lives or behaviors, but both sets of staff indicated that they had experienced other changes that were not listed. Most COVID-19-related experiences were reported at similar rates by staff at baseline and follow up. Worry about being infected, changes in physical health, social distancing, and changes in how staff provided services to clients all slightly decreased from baseline to follow up. The reported loss of childcare, services provided by them being made more difficult, and changes in their mental or physical health all increased from baseline to follow up. To determine if these changes were significant, we conducted paired-samples *t*-tests on staff’s reported affectedness by each item, using Bonferroni correction for multiple analyses. As captured in Table 6, none of the changes from baseline to follow up were significant, indicating that staff also reported the same level of affectedness for these experiences at follow up as they did at baseline.

### 3.4. Qualitative Results (Staff)

The final staff survey question included an open-ended response option for staff to report other changes in their lives or behaviors. These open-ended responses from the baseline and follow-up surveys were reviewed and thematically coded (see Table 7). The coders agreed on 11 primary themes from these responses. The remaining responses that did not fit into these 11 categories included limited food and supplies in stores, online classes, an appreciation for well-being, and struggles in faith.

Although none of the interview questions for staff asked anything specifically about experiences with COVID-19, staff still mentioned difficulties associated with the pandemic during interviews. Across ten interviews with participating staff from the IPV shelter, seven staff explicitly mentioned COVID-19. The primary theme regarding COVID-19-related responses was that work related to the project and in general was made particularly difficult by the pandemic. Staff indicated that COVID-19 restrictions and near daily changes to safety protocols changed how they interacted with clients. Two of the interviewees specifically suggested that COVID-19 affected everyone working in the helping field because “we cannot do in-person contact” as usual. The inability to work face-to-face with clients changed how staff were able to interact with clients. Most staff also indicated that they desired more in-person service to resume.

## 4. Discussion

Our findings are relevant as the effects of COVID-19 continue and the potential for future global health crises rises. We describe survivors’ mental health and substance use changes related to the COVID-19 pandemic as well as provider responses to the pandemic so that service providers can better address these concerns with their clients and better support their staff. Both clients and staff reported multiple affecting experiences related to COVID-19. In fact, of the nearly 100 persons we surveyed, only two individuals indicated that COVID-19 had not affected their lives or behaviors. Both clients and service providers indicated that the unintended consequences of the pandemic and related health regulations included not being able to meet their basic needs for food and supplies in addition to financial challenges and mental and emotional health changes. When preparing for the next public health emergency, it will be important that the plans used in 2020 are revised to better meet individuals’ basic needs.

Many of the effects of COVID-19 reported by survivors of violence, such as contracting COVID-19 and worrying about infection, may have been experienced by anyone during the pandemic. Importantly, these results indicate that survivors of violence were also reporting potentially more serious or retraumatizing outcomes such as increased feelings of loneliness, depression, and abuse. The changes from time 1 to time 2 across these more concerning items were not significant, indicating that the clients surveyed at time 1 reported the same level of affectedness for these experiences as those surveyed at time 2. Due to our limited sample size, we also ran nonparametric tests (Mann–Whitney U-tests) for significance on these measures, which revealed the same conclusions. These conclusions were, namely, that survivors reported an increase in negative mental and physical health outcomes because of COVID-19 and related public health regulations, which did not get better even after nearly half a year. These results suggest that mental health will also be a primary concern during the next public health emergency. Providers, communities, and policy makers must work together to address current workforce shortages in mental health and wellness, and researchers and practitioners should further investigate virtual methods of treatment so that we are better prepared in the future.

Furthermore, clients reported an increase in substance misuse in the early pandemic. Although fewer survivors surveyed in the spring of 2021 reported an increase in their alcohol or tobacco use as a result of COVID-19-related experiences, the changes over time were not significant. The survivors surveyed at time 2 also represent a different, independent set of participants which may have behaved differently at time 1. Worryingly, survivors reported pandemic-related increases in illicit drug use, including marijuana, prescription drug misuse, and methamphetamines, both early in the pandemic (summer 2020) and almost half a year later. Although fewer survivors reported an increase in using illegal substances at time 2, scores relating to survivors’ amount of use did not significantly change over time, indicating that the increase in use was similar for survivors who were using at both time points. As the effects of the COVID-19 pandemic continue, those working with survivors of violence need to be aware of the possibility that substance use is common among survivors and may have increased since 2020. This study draws further attention to the need for multisector and co-located services for survivors of violence that include mental and behavioral healthcare and, specifically, trauma-informed substance use treatment and recovery.

Our correlational analysis suggests that feelings of loneliness, depression, safety, and abuse are positively related. The relationship between these negative mental health outcomes and abuse is not surprising given that previous research has found associations between experiencing IPV and depression, PTSD, anxiety, and dysthymia [32,33]. Changes in mental health seem to co-occur with experiences of violence and all changes remained mostly consistent over time, indicating that survivors may not have experienced relief as COVID-19 restrictions have lapsed. Some of these mental health effects could also be related to changes in service provisions and provider capacities as service providers were also suffering from stress due to the pandemic. The clients’ reports of feeling more abused and less safe may be related to actual increases in violence perpetration and/or the retraumatizing effects of COVID-19-related regulations as described in client interviews. It was indeed some of these findings related to the COVID-19 experiences of survivors that led us to add questions related to COVID-19 to our interviews with survivors. Sharing these results may help future researchers and service providers work together to develop pre-emptive interventions for survivors that combat the tendency for public health measures to retraumatize survivors.

The interview results suggested that COVID-19 may be perceived as mirroring the power and controlling aspects of IPV. We present a substantial overlap between the emotional abuse perpetrated by a controlling partner and the impact of public health guidance and fears of infection from COVID-19 (see Figure 2). Comparable to the bodily harm of physical violence, postponing healthcare due to a fear of infection or acquiring care in an over-burdened healthcare system can result in adverse physical health effects. Power differentials (privilege) between clients and providers are intensified given restrictions on healthcare access and strict protocols for accessing services. Like a violent perpetrator, COVID-19 restrictions limit access to sexual healthcare, safe-sex supplies, and contraception choices. Support system access is limited by ‘healthy at home’ ordinances. Other commonalities include the menacing appearance of PPE and a lack of control of finances with employment disruptions. Retraumatization may further exacerbate the psychological and physical impacts brought on by the pandemic in addition to triggering survivors of IPV. These findings suggest that the Duluth Model of Power and Control [49] may be useful as a framework through which practitioners can understand and more directly address the possibility of re-occurring trauma due to COVID-19 and future pandemics and which offers the following tangible suggestions for service providers [63,64]:Reduce the menacing appearance of PPE by making your name easily readable or attaching a photograph of yourself;Openly share changes in practice due to COVID-19 and communicate an understanding of how the pandemic and new protocols can be retraumatizing;Include the client’s voice when developing a care/services plan to maintain their autonomy;Reduce accessibility barriers by offering telehealth services.

Our results related to staff experiences may help explain some of the loneliness, depression, and safety concerns experienced by clients. Staff also experienced the expected outcomes of living through a pandemic, including a lack of access to basic needs, financial instability, and worry about infection. Staff also experienced negative mental health outcomes and reported that they made changes to how they worked with clients because of COVID-19 experiences. This study confirms that IPV service providers, like others in the helping field, were greatly affected by the pandemic. It is highly likely that staff experiencing such stressors were unable to provide pre-pandemic levels of attentiveness and care, despite their best intentions. Some pre-pandemic services, such as shelter capacity and face-to-face counseling, also decreased or ceased entirely during the pandemic to meet COVID-19-related safety precautions. These changes in quality or perceived quality of care could provide some reasoning as to why survivors were experiencing worsening mental health and seeking other coping mechanisms, such as substance misuse. Other themes that have come out of working and talking with IPV service providers include challenges due to low retention of staff, a lack of hazard pay, and a general lack of workforce available and willing to take on the demanding work involved with supporting survivors of violence. Furthermore, staff desired to see services return to pre-pandemic methods, especially to connect more in person with clients. Charitable and government funders should consider providing additional support and extending incentives so that service providers working with this population can be paid equitably relative to the risks and difficulties of this work.

The primary limitation of this study is its sample size. Our sample of survivors at time 1 was small (*n* = 19) and at time 2 even smaller (*n* = 15). Further, a limitation of all cross-sectional research is the possibility that the differences in individuals participating over time may impact the outcomes being studied. Our sampling method is also subject to sampling and self-selection bias, another limitation of this work. Additionally, interviews were conducted with only six survivors and all data collection was limited to a sample of individuals living in the same Southern state and receiving residential services from the same provider. Our staff sample was larger and enabled a repeated-measures analysis but was still limited in size (*n* = 35) and geographic location and by the fact that all staff were employed by the same service-providing organization. Given the limitations of this study, including sample size and all participants residing in the same shelter, we caution against generalizing these results to the wider population. Future research should be conducted to confirm these findings and test its applicability for other kinds of trauma-informed care.

Our results are also limited by the fact that survivors’ and staff’s changes in mental/physical health and changes in substance misuse by survivors are both self-reported. Future work could confirm physical changes, such as COVID-19 diagnoses, as well as emotional changes by measuring depression and anxiety using validated instruments. Although two of our reference sources for instrument creation have not been validated, they have been used by other researchers and come from reputable sources (i.e., the Harris Poll). Our interviews with clients and staff were also limited by individual availability and the capacity of our partner IPV service-providing organization. Due to the challenges of the pandemic, including a lack of workforce and PPE, staff were unable to recruit and schedule interviews with more than six clients residing in the shelter in the summer of 2020. The turnover of staff at the organization was also very high (at least 44 staff members left the organization between 2020 and 2021), which limited our ability to conduct paired analyses and recruit for staff interviews. Finally, staff interviews were not designed to measure COVID-19-related information. Adding COVID-19-related questions to staff interviews would allow for further investigation of the challenges and changes related to COVID-19.

## 5. Conclusions

COVID-19 has been and continues to be devastating for many reasons. The impacts and ramifications of this pandemic have yet to be fully understood; however, it has also shed light on the compounding effects that trauma like IPV has on individuals in ways not previously recognized. The data reported here show that survivors indeed reported experiencing changes in mental health (feelings of depression, loneliness, abuse, and lowered perceptions of safety) and substance use (changes in tobacco, alcohol, and illicit substance use) related to the pandemic. Our findings are interpreted in the context of qualitative data from in-depth interviews suggesting that COVID-19-related movement restrictions and stay-at-home orders mimic survivors’ experiences of power and control in violent relationships. Community-based organizations, such as violence shelters, can help mitigate the ongoing impacts of COVID-19 and provide an important voice in developing more nuanced public health efforts for survivors while continuing to maintain essential health and safety protocols. Service-providing organizations should also be cautious about adding to the existing workload of their staff, as this study also shows that service providers were adversely affected by the pandemic, including in their ability to provide needed services to survivors of IPV. To our knowledge, the parallels between COVID-19 public health measures and IPV have not been explored, and these findings offer a new theoretical framework to the field. Our results also provide opportunities for systems to engage in greater empathy and multifaceted support to mitigate the impacts of IPV on survivors in their recovery.

## Figures and Tables

**Figure 1 ijerph-20-02896-f001:**
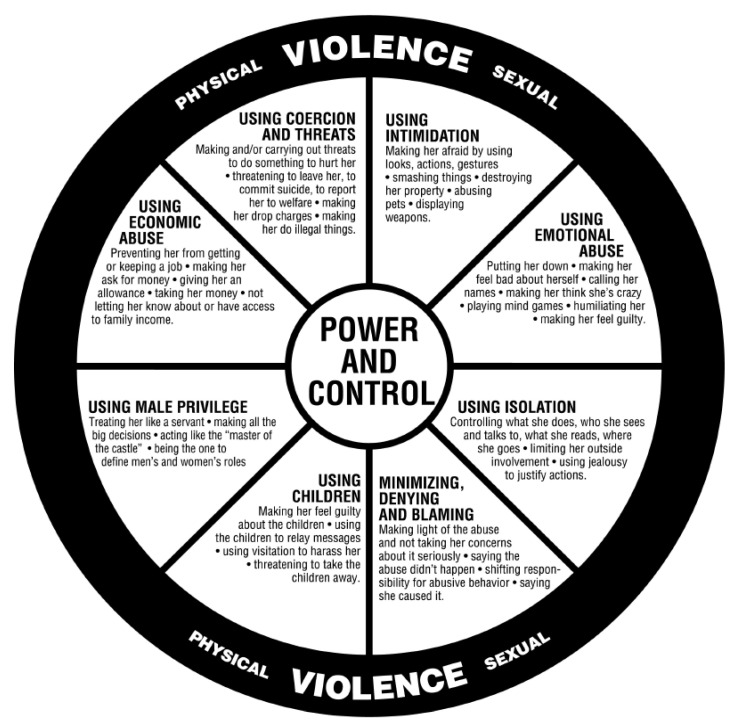
The Duluth Models Power and Control Wheel [49]. Reprinted with written permission from Domestic Abuse Intervention Programs (DAIP) of Duluth, MN.

**Figure 2 ijerph-20-02896-f002:**
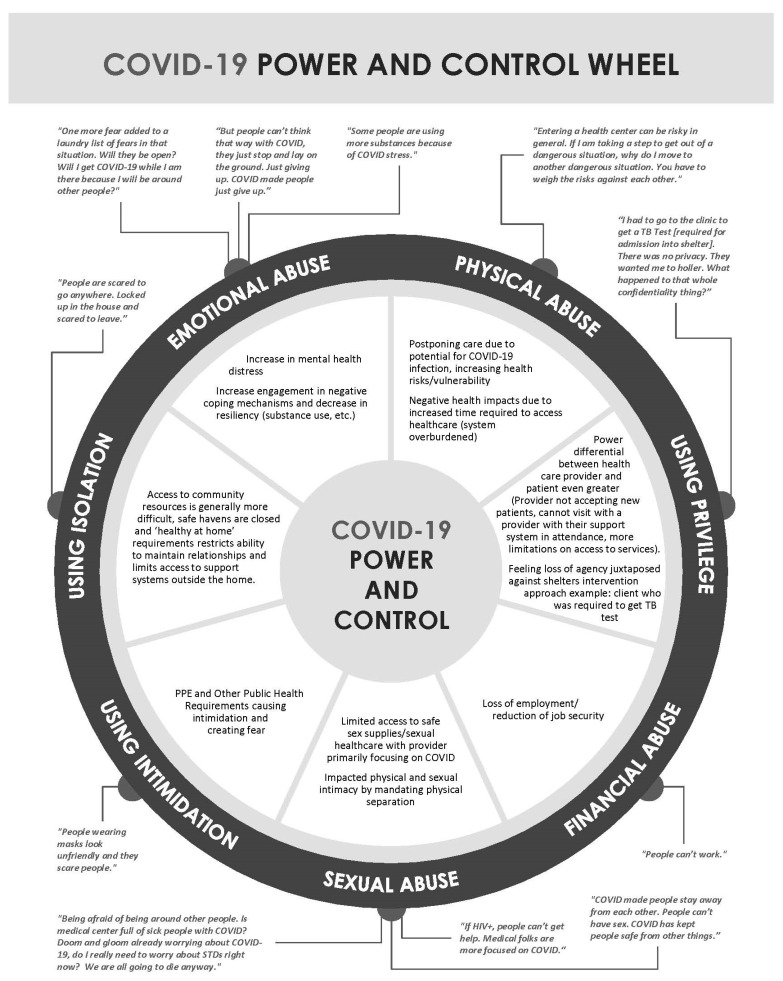
COVID-19 power and control wheel. Duluth’s Power and Control Wheel [49] was used as a framework to show how COVID-19 emphasized and provided new examples of IPV. This adaptation was first presented by Bauer et al. [50] at the 14th Annual InWomen’s Conference.

**Table 1 ijerph-20-02896-t001:** Clients’ demographic characteristics.

Items						
	Time 1 (*n* = 19)			Time 2 (*n* = 15)		
	Minimum	Maximum	Mean	Minimum	Maximum	Mean
Age	21	61	38.8	29	57	38.3
Number of Children	0	4	1.87	1	5	2.80
**Items**	**Response Frequencies**					
	**Time 1**			**Time 2**		
**Relationship Status**	** *n* **	**%**		** *n* **	**%**	
Married	3	15.8		1	7.1	
Dating > 1 person	0	--		1	7.1	
Dating 1 person	0	--		2	14.3	
Dating 1 person in a serious relationship	1	5.3		3	21.14	
Single	15	78.9		7	50.0	
Total	19	100.0		14	100.0	
**Gender Identity**	** *n* **	**%**		** *n* **	**%**	
Female	16	84.2		12	80.0	
Male	1	5.3		2	13.3	
Other (transgender, other, or preferred not to answer)	2	10.6		1	6.7	
Total	19	100.0		15	100.0	
**Highest Level of Education**	** *n* **	**%**		** *n* **	**%**	
Some college	8	42.1		5	33.3	
Some high school	4	21.1		4	26.7	
College degree	0	--		3	20.0	
High-school graduate or GED	4	21.1		2	13.3	
Less than high school	3	15.8		1	6.7	
Total	19	100.0		15	100.0	
**Health Insurance Status**	** *n* **	**%**		** *n* **	**%**	
No health insurance	2	10.5		1	6.7	
Private insurance	2	10.5		1	6.7	
Public insurance (Medicaid, Medicare)	14	73.7		13	86.7	
Total	18	100.0		15	100.0	
**Employment Status**	** *n* **	**%**		** *n* **	**%**	
Not currently working	13	68.4		13	86.7	
Working full or part time or a student	5	26.4		2	13.3	
Total	18	100.0		15	100.0	
**Race**	** *n* **	**%**		** *n* **	**%**	
White/Caucasian	9	47.4		8	53.3	
Black or African American	5	26.3		5	33.3	
American Indian	4	21.2		3	20.0	
Other ^a^	9	47.4		2	6.7	
Total	18	100.0		15	100.0	

^a^ American Indian; American Indian, White/Caucasian; Black or African American, Native Hawaiian or Other Pacific Islander; Bosnian/Eastern European.

**Table 2 ijerph-20-02896-t002:** Clients’ COVID-19 experiences.

Experiences	Response Frequencies				Level of Affectedness				Significance	
	Time 1 (*n* = 19)		Time 2 (*n* = 15)		Time 1 (*n* = 19)		Time 2 (*n* = 15)			
	*n*	%	*n*	%	*n*	Mean ^a^	*n*	Mean ^a^	(df) *t*	*p*
COVID-19 diagnosis	13	68.4	0	0	13	0.15	0	0	--	--
Worry/anxiety about being infected	18	94.7	5	33.3	17	1.88	8	1.83	(27) 0.08	0.93
Changes in physical health	13	68.4	5	33.3	12	0.75	6	0.91	(21) −0.34	0.74
Not having enough basic supplies	16	84.2	6	40.0	15	2.47	8	2.09	(24) 0.57 ^b^	0.58
Children experiencing distress	14	73.7	5	33.3	14	1.5	4	1.33	(21) 0.27 ^b^	0.79
Interpersonal conflict with family or loved ones	18	94.7	8	53.3	16	2.38	8	2	(25) 0.62	0.54
Feeling more depression	19	100.0	9	60.0	17	2.65	10	2.42	(27) 0.45	0.66
Feeling more loneliness	19	100.0	9	60.0	17	2.76	10	2.5	(27) 0.47 ^b^	0.65
Feeling less safe	17	89.5	9	60.0	15	2.27	10	2.31	(26) −0.07	0.95
Feeling more abused	17	89.5	10	66.7	16	2.25	10	2.83	(26) −1.07	0.29
No changes to my life or behavior	0	0	2	13.3	--	--	--	--	--	--
**Coping Strategies**	** *n* **	**%**	** *n* **	**%**	** *n* **	**Mean ^a^**	** *n* **	**Mean ^a^**	**(df) *t ^c^***	** *p* **
Feeling grateful for the break from usual activities	12	63.2	13	86.7	12	0.54	5	1.0	(23) 0.92	0.37
Connecting with others, including talking with people you trust about your concerns and how you are feeling	15	78.9	10	66.7	14	1.17	9	1.25	(24) 1.37	0.18
Needing more/less sleep or other changes to your normal sleep pattern	16	84.2	8	53.3	14	2.54	11	1.5	(25) −0.44	0.67
Taking breaks from watching, reading, or listening to news stories, including social media	16	84.2	9	60.0	15	3.0	10	1.4	(25) −2.65	0.01
Healthy behaviors like trying to eat healthy, well-balanced meals, exercising regularly, getting plenty of sleep, or avoiding alcohol and drugs	14	73.7	10	66.7	14	2.0	10	1.75	(24) 0.21	0.84
Engaging in more family activities	12	63.2	13	86.7	12	1.55	6	1.5	(21) −0.34	0.74
Talking with a mental health care provider.	17	89.5	8	53.3	15	1.92	10	1.5	(25) 0.14	0.89

^a^ Scale: 0 = not at all to 4 = greatly/a lot. ^b^ Where indicated, Levene’s test for equal variances was significant, and reported *t*-test and significance levels reflect an assumption for unequal variances. ^c^ Equal variances assumed for all items.

**Table 3 ijerph-20-02896-t003:** Clients’ substance use.

Substance (Change in Use)	Response Frequencies				Change in Use				Significance	
	Time 1 (*n* = 19)		Time 2 (*n* = 15)		Time 1 (*n* = 19)		Time 2 (*n* = 15)			
	*n*	%	*n*	%	*n*	Mean ^e^	*n*	Mean ^e^	(df) *t* ^f^	*p*
Alcohol	17	89.5	11	73.3	10	3.56	5	2.90	(13) −0.37	0.72
Illicit or illegal substances (e.g., marijuana, prescription drugs not prescribed to you, or meth).	11	57.9	4	26.7	12	3.10	4	3.40	(14) 0.96	0.32
Tobacco use (e.g., smoking cigarettes, e-cigs, vapes, or chewing tobacco).	13	68.4	5	33.3	16	3.58	10	2.75	(24) 1.26	0.22

^e^ Scale: 1 = went down a lot, 2 = went down slightly, 3 = stayed the same, 4 = went up slightly, and 5 = went up a lot; Equal variances assumed for all items. ^f^ Levene’s test for equal variances was significant, and reported *t*-test and significance levels reflect an assumption for unequal variances.

**Table 4 ijerph-20-02896-t004:** Correlations between clients’ experiences.

	Time 1 (*n* = 19)				Time 2 (*n* = 15)			
Clients’ Experience		*r*	*p*	*n*		*r*	*p*	*n*
Depression	Loneliness *	0.697	0.002	17	Loneliness *	0.924	<0.001	12
	Less safe	0.508	0.053	15	Less safe *	0.712	0.009	12
	More abused	0.561	0.029	15	More abused *	0.74	0.006	12
	Tobacco use	0.216	0.458	14	Tobacco use	0.411	0.272	9
	Alcohol use	0.557	0.152	8	Alcohol use	0.738	0.262	4
	Illicit use	0.269	0.451	10	Illicit use	0	1	3
Loneliness	Less safe	0.493	0.062	15	Less safe	0.668	0.018	12
	More abused *	0.717	0.003	15	More abused	0.508	0.092	12
	Tobacco use	0.085	0.772	14	Tobacco use	0.32	0.402	9
	Alcohol use	0.6	0.115	8	Alcohol use	0.258	0.742	4
	Illicit use	−0.073	0.842	10	Illicit use	−0.5	0.667	3
Less safe	More abused	0.449	0.093	15	More abused	0.47	0.123	12
	Tobacco use	0.099	0.749	13	Tobacco use	0.41	0.273	9
	Alcohol use	0.478	0.231	8	Alcohol use	0.703	0.185	5
	Illicit use	−0.453	0.188	10	Illicit use	0	1	4
More abused	Tobacco use	0.022	0.941	14	Tobacco use	0.187	0.63	9
	Alcohol use	0	1	9	Alcohol use	0.775	0.225	4
	Illicit use	−0.393	0.231	11	Illicit use	1.000		3

* Denotes significance.

**Table 5 ijerph-20-02896-t005:** Staff’s demographics.

Items	Baseline (*n* = 59)			Follow Up (*n* = 40)		
	Minimum	Maximum	Mean	Minimum	Maximum	Mean
Age	21	61	38.8	29	57	38.3
**Professional Experience**						
Months worked at the center	2	300	46.72	1	228	47.7
Months worked with people who have experienced IPV	0	300	54.72	0	420	79.0
**Items**	**Response Frequencies**					
	**Baseline (*n* = 59)**			**Follow Up (*n* = 40)**		
**Gender Identity**	** *n* **		**%**	** *n* **		**%**
Female	46		79.3	31		83.8
Male	6		10.3	4		10.8
Nonbinary/prefer not to say	6		3.4	1		2.7
Total	58		100.0	37		100.0
**Highest Level of Education**	** *n* **		**%**	** *n* **		**%**
High-school graduate or GED	1		1.7	1		2.7
Some college	5		8.6	1		2.7
College degree	34		58.6	23		62.2
Post-graduate degree	18		31	12		32.4
Total	58		100.0	37		100.0
**Current Job Title**	** *n* **		**%**	** *n* **		**%**
Advocate	28		49.1	23		62.2
Administrative positions	13		22.8	4		10.8
Supervisor	4		7.0	4		10.8
Counselor/Therapist	3		5.3	2		5.4
Prevention Coordinator	2		3.5	--		--
Director	2		3.5	2		5.4
Other ^a^	5		8.8	2		5.4
Total	57		100.0	37		100.0
**Race/Ethnicity**	** *n* **		**%**	** *n* **		**%**
Hispanic/Latino	2		3.5	0		0
White/Caucasian	42		73.7	25		62.5
Black or African American	9		15.8	11		27.5
Asian, Native Hawaiian, or other Pacific Islander	1		1.8	1		2.5
American Indian	1		1.8	0		0
Other	2		3.4	1		2.5
Total	57		100.0	38		100.0

^a^ Intern Therapist/Coordinator, Volunteer Coordinator, Art Therapist, Building Services, Coordinator, Outreach Coordinator, DV Housing Coordinator.

**Table 6 ijerph-20-02896-t006:** Staff’s COVID-19 experiences.

Experience	Response Frequencies				Average Affected Score				Significance ^b^ (*n* = 35)	
	Baseline (*n* = 59)		Follow Up (*n* = 40)		Baseline (*n* = 59)		Follow Up (*n* = 40)			
	*n*	%	*n*	%	*n*	Mean ^a^	*n*	Mean ^a^	(df) *t*	*p*
Worry/anxiety about being infected	49	83.1	32	80.0	49	2.86	31	2.74	(20) −0.33	0.75
Changes in physical health	19	32.2	12	30.0	18	2.44	12	2.75	(5) 0.00	1.00
**Challenges in your workplace due to:**	** *n* **	**%**	** *n* **	**%**	** *n* **	**Mean ^a^**	** *n* **	**Mean ^a^**	**(df) *t***	** *p* **
Social distancing	44	74.6	29	72.5	44	3.00	28	2.79	(14) 0.52	0.61
Loss of childcare	7	11.9	5	12.5	7	3.29	5	2.20	(1) −0.33	0.80
Technology issues (e.g., loss of internet)	17	28.8	12	30.0	17	2.24	12	2.08	(4) 0.54	0.62
Services you provide were made difficult by the precautions	32	54.2	22	55.0	32	2.66	21	2.57	(9) −0.56	0.60
Changes in client mental and/or physical health	26	44.1	27	67.5	26	2.88	26	3.00	(11) −2.70	0.02
Changes in how you provide services to your clients	42	71.2	24	60.0	42	2.95	23	2.74	(15) −0.22	0.83
If yes, how confident do you feel in administering services this new way?	--	--	--	--	40	2.90	23	2.83	(14) 1.83	0.09
None of the above	2	3.4	0	0	--	--	--	--	--	--
Have there been other changes to your life or behavior due to COVID-19?	36	61.0	21	55.3	--	--	--	--	--	--

^a^ Scale: 0 = not at all to 4 = greatly/a lot. ^b^ Significance tests represent paired *t*-tests.

**Table 7 ijerph-20-02896-t007:** Staff’s open-ended responses of COVID-19 experiences.

Themes	Response Frequencies			
	Baseline (*n* = 36; 75 Responses)		Follow Up (*n* = 21; 36 Responses)	
	*n*	%	*n*	%
Changes to spending time with family or friends and/or challenges with not seeing or being physically close to family or friends	15	20.0	5	13.9
Changes to self-care plans/coping strategies, including hobbies and/or recreational activities	13	17.3	3	8.3
Mental health challenges and/or emotional distress	12	16.0	12	33.3
Mental health challenges and/or emotional distress specifically due to work changes or challenges	5	6.7	4	11.1
Work changes and challenges (e.g., longer hours, staff capacity, or telecommuting)	11	14.7	6	16.7
Concerns and/or worries about oneself or others being infected, including increased safety precautions	8	10.7	4	11.1
Changes in schedules, routines, or roles	4	5.3	--	--
Competing professional and parental responsibilities, including child’s schoolwork; difficulty with time management	3	4.0	2	5.6
Difficulty planning for future	2	2.7	--	--
Personal loss	2	2.7	3	8.3
Financial challenges and/or budgeting changes	2	2.7	1	2.8
Other	3	4.0	2	5.6

## Data Availability

The data presented in this study are available on request from the corresponding author. The data are not publicly available due to confidentiality concerns for IPV staff and survivors of violence.

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
