# Peer review of "Changes in Mental Health, Emotional Distress, and Substance Use Affecting Women Experiencing Violence and Their Service Providers during COVID-19 in a U.S. Southern State"

_ijerph, 2023, doi:10.3390/ijerph20042896_

Round 1
Reviewer 1 Report
Abstract
Line 10: Order of writing: I think that the goal of this study should be written first, followed by the most important results. I also think that the type of study should be written (the type of study should be repeated in the material and methods)
Line 19: I suggest to replace the term negative, for example, the word deterioration or similar
Line 24: Briefly, how it had a adversely impact
Line 51: Use the International Classification of Diseases (ICD) code in parentheses next to the name of each disease
Line 105: Is there information in the mentioned references about how many care providers left the workplace in order to estimate the size of the problem
Line 107: for example, what kind of resources, do you mean resources in the next sentence (food, health care,... professional support,...)
Line 129: I suggest explicite to write a study design (like cohort or cross-sectional study??...) and sampling method
Line 400: I suggest summarizing the same types of data for better visibility of the table. I think it makes no sense to show the parameter eg gender in four forms, because other forms of gender can hardly be used for any analysis, so they can be summarized. Do the same with, for example, employment
Line 479: Figure 1 was not created from the data collected and processed for the purposes of this article, so it has no place in the results but in the introduction, possibly in the methods. Also, I don't know why this image would be used at all when the context of the discussion is not about that image. Maybe the discussion should be about the data from that picture!
Line 548: In general, to make it easier to follow the discussion, it is necessary to write down which table each part of the discussion refers to. Likewise, I believe that something that is not significant should not even be mentioned because it is unnecessary. Also, it is difficult to follow the discussion because you need a lot of time to find the data that is discussed in the results, so it needs to be supplemented by adding the table number in parentheses
Line 573: In which table are these results presented and why was the difference not presented (quantified) at this point in the discussion, to make it easier to follow the conclusions. Are the behaviors listed in the sentences that follow in this paragraph the cause of such a result?
Line 639: This comment depends on the study design and sampling method, which is not explicitly written in the article in the material and methods chapter. I believe that this sentence should be written only if the criteria of representativeness of the sample are met
Line 646: I think this sentence is redundant. It raises the question of why such tests were not carried out in this study
Line 657: The conclusion should refer only to significant results as well as the possible recommendations that may arise from these results
Author Response
Response to Reviewers
We have addressed all the reviewer comments using italicized text in the response to reviewers below.
Reviewer 1
Abstract Line 10: Order of writing: I think that the goal of this study should be written first, followed by the most important results. I also think that the type of study should be written (the type of study should be repeated in the material and methods)
We have re-organized the abstract to meet these qualifications by putting the aim of the study first, including the methods used, and placing the results immediately following the aim. We thank Review 1 for this comment as we believe the abstract is now stronger.
Line 19: I suggest to replace the term negative, for example, the word deterioration or similar Line 24: Briefly, how it had a adversely impact
We have changed “negative” to “mental health decline” and “adversely affected” to “experienced mental and emotional stress” in an attempt to meet this request for clarity.
Line 51: Use the International Classification of Diseases (ICD) code in parentheses next to the name of each disease
Although we appreciate this suggestion, we have not added the ICD codes to this list. Rather than specific diagnoses, this list includes broad conditions like “smoking” which would constitute multiple ICD codes (e.g., for smoking, F17.2, 099.33, T65.2, Z72, and Z87.8). Listing out all of the codes for each condition in the list would be unnecessarily long and our understanding is that this is not a journal requirement.
Line 105: Is there information in the mentioned references about how many care providers left the workplace in order to estimate the size of the problem
This is a very important suggestion by Reviewer 1, but, unfortunately, one that we cannot address directly. None of our current references mention the amount of workforce lost to COVID-19. We were able to find resources suggesting that 30-50% of the health care providing workforce has left their work. However, estimates for IPV-specific service providers were not found. Moreover, we only found one peer-reviewed estimate (and two popular press mentions) of workforce losses:
- Sinsky, C. A., Brown, R. L., Stillman, M. J., & Linzer, M. (2021). COVID-related stress and work intentions in a sample of US health care workers. Mayo Clinic Proceedings: Innovations, Quality & Outcomes, 5(6), 1165-1173.
- https://www.pcpcc.org/2022/01/07/nearly-1-5-health-care-workers-have-quit-their-jobs-during-pandemic
- https://www.forbes.com/sites/jackkelly/2022/04/19/new-survey-shows-that-up-to-47-of-us-healthcare-workers-plan-to-leave-their-positions-by-2025/
We do mention the number of staff who left our partner organization in the discussion.
Line 107: for example, what kind of resources do you mean resources in the next sentence (food, health care,... professional support,...)
With apologies, the authors do not understand this comment. Lines 105-109 read, “Generally, providers labeled as essential workers experienced greater stress, a lack of access to necessary resources (personal protective equipment), and greater burnout during COVID-19 than before.40, 49-51 Many social service providers indicated they were unprepared for the professional challenges of COVID-1950 and were impacted personally by primary pandemic consequences such as not enough access to food or healthcare and becoming ill as well as secondary consequences such as stress.” An example of resources is provided “PPE,” and the next sentence describes some limiting factors such as access to food and healthcare. We are unsure of how to address this comment.
Line 129: I suggest explicit to write a study design (like cohort or cross-sectional study??...) and sampling method
We thank reviewer 1 for this suggestion, the cross-sectional and repeated measures design notes have been added throughout the text. We also changed the presentation of the two survivor surveys to “time 1” and “time 2” rather than “baseline” and “follow-up” in the hopes that this change makes the study design clearer.
Line 400: I suggest summarizing the same types of data for better visibility of the table. I think it makes no sense to show the parameter eg gender in four forms, because other forms of gender can hardly be used for any analysis, so they can be summarized. Do the same with, for example, employment
Thank you. We have merged the suggested data types in this table and in the staff demographic table (5).
Line 479: Figure 1 was not created from the data collected and processed for the purposes of this article, so it has no place in the results but in the introduction, possibly in the methods. Also, I don't know why this image would be used at all when the context of the discussion is not about that image. Maybe the discussion should be about the data from that picture!
To clarify for reviewer 1, the majority of Figure 1 was indeed created from the data collected as a part of this article and we have added text to make that clearer. The quotes presented on the outside of the “wheel” and the description of different kinds of abuse inside the “spokes” are all new contributions from this study. We would also like to point the reviewer to revised lines 611-634 where this figure is referenced in the discussion.
Line 548: In general, to make it easier to follow the discussion, it is necessary to write down which table each part of the discussion refers to. Likewise, I believe that something that is not significant should not even be mentioned because it is unnecessary. Also, it is difficult to follow the discussion because you need a lot of time to find the data that is discussed in the results, so it needs to be supplemented by adding the table number in parentheses
Thank you, again. We have added parenthetical references to each table where data referenced in the discussion can be found. We have left the discussion of non-significant changes in mental health because it shows that survivors’ substance use and mental health concerns early in the pandemic remained unchanged 8-months later. (From the text, “The changes from time 1 to time 2 across these more concerning items were not significant, indicating that the clients surveyed at baseline reported the same level of affectedness for these experiences as those surveyed at time 2.”)
Line 573: In which table are these results presented and why was the difference not presented (quantified) at this point in the discussion, to make it easier to follow the conclusions. Are the behaviors listed in the sentences that follow in this paragraph the cause of such a result?
As above, we thank the reviewer for their suggestion to add table references and have done so here as well. We are unsure of exactly what else the reviewer is referring to with this statement. Lines 570-574 read, “Although fewer survivors surveyed in Spring of 2021 reported increases in their alcohol or tobacco use as a result of COVID-19 related experiences, the changes over time were not significant. The survivors surveyed at follow-up also represent a different, independent set of participants which may have behaved differently at baseline as well.” The first sentence corresponds to non-significant changes in alcohol and tobacco use that are reported in the results section line 431-438. The difference in the survivor sample from time one to time two is reported in the results lines 391-392. We hope we have addressed the reviewer’s concern by adding the text to the methods section (“Each recruitment period included a completely distinct set of individuals as those living in residential shelter changed between 2020 and 2021”) and the discussion (“Further, a limitation of all cross-sectional research is the possibility that the differences in individuals participating over time may impact the outcomes being studied.”)
Line 639: This comment depends on the study design and sampling method, which is not explicitly written in the article in the material and methods chapter. I believe that this sentence should be written only if the criteria of representativeness of the sample are met
We have, as the reviewer suggests, added mention of the repeated cross-sectional design throughout the text. Furthermore, while recruitment is described in detail, we also clarified that the sampling method was by convenience and voluntary response sampling. We still believe that it is important for “future research should be conducted to confirm these findings and test its applicability for other kinds of trauma-informed care.” But we have also added that “convenience and voluntary response sampling can introduce sampling and self-selection bias” as a limitation to our research.
Line 646: I think this sentence is redundant. It raises the question of why such tests were not carried out in this study
Thank you; we have deleted this sentence.
Line 657: The conclusion should refer only to significant results as well as the possible recommendations that may arise from these results
If the reviewer is referencing the entire conclusion, we are unclear as to what needs to be changed; we do not discuss non-significant findings in the conclusion. We include the sentence, “the data reported here show that survivors indeed reported experiencing changes in mental health (feelings of depression, loneliness, abuse, and lowered perceptions of safety) and substance use (changes in tobacco, alcohol, and illicit substances) related to the pandemic.” as a way of describing overall results.
Reviewer 2 Report
This manuscript, titled “Changes in mental health, emotional distress, and substance use affecting women experiencing violence and their service providers during COVID-19 in a U.S. southern state,” examined the changes in mental health conditions and substance in intimate partner violence (IPV) survivors and IPV support providers. The authors adopted a correlational approach in assessing the mental health and substance use among IPV survivors and support providers at the start of the COVID-19 pandemic and at a one-year follow-up. They also adopted a qualitative approach to understanding how COVID-19-related restrictions reminded survivors of power differential and control. The study and its findings carry important theoretical and practical implications. However, there are several issues that need to be addressed.
· The quantitative component did not use validated questionnaires, and therefore the validity of the findings is questionable. I would suggest removing the quantitative component altogether and focusing only on the qualitative component.
· The study focused on IPV survivors and support providers. While the rationale for each focus was clearly elaborated, it is still unclear how these two foci could be linked together. The authors are advised either to justify why the two foci were studied in the same study or to remove the part related to IPV support providers.
· Duluth’s Model Power and Control Wheel (Pence & Paymer, 1993) was introduced in a rather abrupt way. It would be helpful to introduce it in the Introduction section.
· The practical implications of the findings need more in-depth discussions.
Author Response
Response to Reviewer
We have addressed all the reviewer comments using italicized text in the response to reviewers below.
Reviewer 2
This manuscript, titled “Changes in mental health, emotional distress, and substance use affecting women experiencing violence and their service providers during COVID-19 in a U.S. southern state,” examined the changes in mental health conditions and substance in intimate partner violence (IPV)survivors and IPV support providers. The authors adopted a correlational approach in assessing the mental health and substance use among IPV survivors and support providers at the start of the COVID-19 pandemic and at a one-year follow-up. They also adopted a qualitative approach to understanding how COVID-19-related restrictions reminded survivors of power differential and control. The study and its findings carry important theoretical and practical implications.
Thank you for these comments!
However, there are several issues that need to be addressed. The quantitative component did not use validated questionnaires, and therefore the validity of the findings is questionable. I would suggest removing the quantitative component altogether and focusing only on the qualitative component.
Respectfully, we disagree with this reviewer’s comment and would like to keep the quantitative results in the study report. To provide context, all but two of the instruments used to create our surveys are validated and/or available in the Phenx Toolkit. The Phenx Toolkit is available online for support in determining instrument validity. The two non-validated, non-Phenx instruments include one tool that is freely available for multiple researchers to use online and has been translated into 5 different languages and cited by 84 other authors. The second, the Harris Poll, comes from an organization that the APA contracted with to track COVID-19 related stress. Although it is true that these instruments have not been validated, we believe that the review of all instruments led to the creation of a comprehensive survey to address changes in survivor and provider physical and mental health resulting from the pandemic. We have also added this as a limitation in the discussion, “Although two of our reference sources for instrument creation have not been validated, they have been used by other researchers and come from reputable sources (i.e., the Harris Poll).”
The study focused on IPV survivors and support providers. While the rationale for each focus was clearly elaborated, it is still unclear how these two foci could be linked together. The authors are advised either to justify why the two foci were studied in the same study or to remove the part related to IPV support providers.
Thank you. We have added text to the introduction and discussion to connect the survivor and provider results.
Duluth’s Model Power and Control Wheel (Pence &Paymer, 1993) was introduced in a rather abrupt way. It would be helpful to introduce it in the Introduction section.
We agree and have added this to the introduction.
The practical implications of the findings need more in-depth discussions.
We have added many more specific calls for practical solutions to the implications of our results and hope that these changes meet the reviewer’s suggestion.
Reviewer 2 also noted that English language and style are fine/minor spell check required, however, did not indicated where there were grammar or spelling errors. We have edited our manuscript and believe that there are no remaining errors.
Reviewer 3 Report
ABSTRACT
The research topic is of scientific and social interest.
There is cohesion in the article, between the section on Theoretical Framework and the subsequent sections, which describe the study and draw conclusions, have a correct, well-structured and cohesive design.
In general, the article is correct and I consider that the topic is in line with the journal’s research objectives.
INTRODUCTION:
The study objective is well defined and identified in both the abstract and the introduction.
The subject under investigation is of growing scientific and social interest. The investigation is current.
The result is a work that can be the basis for many others in this field.
The statistical technique used is well justified and explained, both the process and the results, despite not being widely used in these investigations, which implies an added effort by the authors of the article.
MATERIALS, METHODS and RESULTS:
The statistical treatment is correct.
DISCUSSION and CONCLUSION:
The conclusions are well drawn and interesting.
The discussion is correct.
Author Response
Response to reviewers
We have addressed all the reviewer comments using italicized text in the response to reviewers below.
Reviewer 3
ABSTRACT The research topic is of scientific and social interest. There is cohesion in the article, between the section on Theoretical Framework and the subsequent sections, which describe the study and draw conclusions, have a correct, well-structured and cohesive design. In general, the article is correct, and I consider that the topic is in line with the journal’s research objectives.
INTRODUCTION: The study objective is well defined and identified in both the abstract and the introduction. The subject under investigation is of growing scientific and social interest. The investigation is current. The result is a work that can be the basis for many others in this field. The statistical technique used is well justified and explained, both the process and the results, despite not being widely used in these investigations, which implies an added effort by the authors of the article.
MATERIALS, METHODS and RESULTS: The statistical treatment is correct.
DISCUSSION and CONCLUSION: The conclusions are well drawn and interesting. The discussion is correct
Author Response:
We appreciate Reviewer 3’s comments. We have not made direct edits to the text as Reviewer 3 did not indicate that any edits needed to be made. However, we do want to appreciate and acknowledge that Reviewer 3 is in opposition to Reviewer 2 by saying that our statistical treatment is correct, and the conclusions are well drawn.
Round 2
Reviewer 2 Report
Thanks for addressing the comments.
Author Response
We would like to thank Reviewer 2 for their second review. It does not appear that Reviewer 2 has any specific remaining concerns, aside from the research design, which cannot be changed at this time. We have thoroughly checked the manuscript for errors.